cognition

body size, voice, acoustic communication, nonlinear vocal phenomena, pitch, roughness

**Author for correspondence:**
Andrey Anikin
e-mail: andrey.anikin@lucs.lu.se

# Harsh is large: nonlinear vocal phenomena lower voice pitch and exaggerate body size

Andrey Anikin[1,2], Katarzyna Pisanski[2,3], Mathilde Massenet[2] and David Reby[2]

[1]Division of Cognitive Science, Lund University, 22100 Lund, Sweden
[2]Equipe de Neuro-Ethologie Sensorielle, CNRS and University of Saint Étienne, UMR 5293, 42023 St-Étienne, France
[3]CNRS, French National Centre for Scientific Research, Laboratoire de Dynamique du Langage, University of Lyon 2, 69007 Lyon, France

  AA, 0000-0002-1250-8261; KP, 0000-0003-0992-2477; MM, 0000-0002-0085-1871; DR, 0000-0001-9261-1711

A lion's roar, a dog's bark, an angry yell in a pub brawl: what do these vocalizations have in common? They all sound harsh due to nonlinear vocal phenomena (NLP)—deviations from regular voice production, hypothesized to lower perceived voice pitch and thereby exaggerate the apparent body size of the vocalizer. To test this yet uncorroborated hypothesis, we synthesized human nonverbal vocalizations, such as roars, groans and screams, with and without NLP (amplitude modulation, subharmonics and chaos). We then measured their effects on nearly 700 listeners' perceptions of three psychoacoustic (pitch, timbre, roughness) and three ecological (body size, formidability, aggression) characteristics. In an explicit rating task, all NLP lowered perceived voice pitch, increased voice darkness and roughness, and caused vocalizers to sound larger, more formidable and more aggressive. Key results were replicated in an implicit associations test, suggesting that the 'harsh is large' bias will arise in ecologically relevant confrontational contexts that involve a rapid, and largely implicit, evaluation of the opponent's size. In sum, nonlinearities in human vocalizations can flexibly communicate both formidability and intention to attack, suggesting they are not a mere byproduct of loud vocalizing, but rather an informative acoustic signal well suited for intimidating potential opponents.

## 1. Introduction

Differences in relative body size are of crucial importance for social animals, and displays of size and dominance are among the most common and important messages conveyed by animal signals [1,2]. In a constant evolutionary arms race, receivers benefit from attending to indicators of body size and fighting ability, with selection often favouring signallers who appear as large and formidable as possible [2], with humans appearing as no exception [3,4]. As a result, many mammals have evolved anatomical adaptations for exaggerating their size. In the vocal domain, such adaptations include vocal sacs and descended or movable larynges in a number of terrestrial mammals, which serve to enlarge the acoustic resonator and therefore lower the resonance frequencies (formants) [5,6].

In humans, in addition to a permanently descended larynx [7], men and women in diverse cultures can effectively exaggerate their size by volitionally lowering their vocal tract resonances (formants) as well as their fundamental frequency ($f_0$, perceived as voice pitch) [3,8]. In turn, human listeners systematically associate low voice pitch with a large size and physical prowess [9,10]. This 'low is large' perceptual bias is often incorrect in the sense that there is at best a weak correlation between $f_0$ and the size of the vocalizer within sex and age groups [11]. Nevertheless, the perceptual association is incredibly strong [12]. Many other animals have also evolved specific anatomical

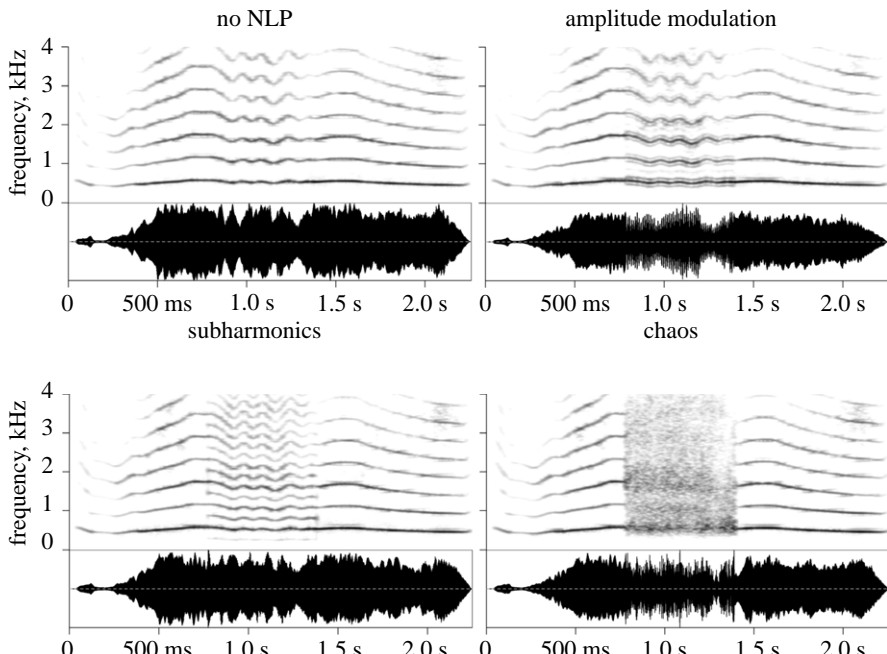

**Figure 1.** Spectrograms of a prototype vocalization (female cry, file 'cry_F_475') resynthesized with different nonlinear vocal phenomena (NLP). Amplitude modulation is created by multiplying the signal by a relatively low-frequency waveform ($M \pm$ s.d. $= 90 \pm 20$ Hz); subharmonics are synthesized as a new, harmonically related voiced component at 1/2, 1/3 or 1/4 of the original fundamental frequency; chaos is emulated by adding strong jitter—rapid random pitch changes (audio and code for creating the stimul on http://cogsci.se/publications.html).

adaptations to lower $f_0$ such as fleshy pads on the vocal folds of the big cats, hypertrophied larynges in howler and colobus monkeys and additional non-laryngeal vocal folds in koalas (reviewed in [5]), suggesting that the 'low pitch sounds large' association may be pervasive across mammalian species [13]. More generally, the way to sound big may to be to produce loud [14,15] vocal sounds with prominent low-frequency components [16]. In this study, we provide evidence that other voice frequency components, apart from the extensively studied $f_0$ and formants, are also involved in acoustic size exaggeration. Namely, we show that vocal harshness caused by the presence of nonlinear vocal phenomena (NLP) can function to exaggerate body size in human vocalizations from roars to screams.

## (a) Why would vocal harshness signal largeness and threat?

In a tonal sound, such as a sustained vowel produced with a modal voice register, the vibrating vocal folds produce a single carrier frequency $f_0$, some harmonics of which are amplified by the resonances of the vocal tract (formant frequencies) [7]. However, the vibratory regimes of the vocal folds can often be more complex and irregular, producing a variety of so-called NLP. The most commonly reported types of NLP include frequency jumps, sidebands or amplitude modulation, subharmonics and deterministic chaos [17]. All these NLP are common in the calls of a large variety of mammals [17–21], birds [22] and even amphibians [23,24]. While comparatively much less studied in humans, NLP have been documented in the cries of human infants [25,26] and in aversive (e.g. pain and fear) vocalizations of adult humans [27,28].

NLP create an irregular or harsh voice quality and convey important information about caller and context. They are often found in loud, high-arousal calls [17,29,30] that attract the listeners' attention [28] and, due to their inherent

irregularity, prevent habituation [31,32]. When manipulated experimentally in human vocal stimuli, NLP enhance the perceived emotion intensity of a vocalization, particularly the intensity of negatively valenced emotions such as aggression [33,34]. This agrees well with Morton's motivation–structural rules, according to which low and rough voices signal aggression across animal species [13,35]. What might be the mechanism behind this effect? Is it the rough voice quality associated with NLP that makes them a robust vehicle for expressing aggression, or might NLP somehow lower the perceived frequency of a vocalization, contributing to the impression of larger body size?

In a seminal article on the role of NLP in animal acoustic communication, Fitch and colleagues [17] hypothesized that subharmonics, a type of NLP, should be common in aggressive animal vocalizations because they lower perceived pitch and thereby exaggerate the caller's size. Subharmonics appear on the voice spectrogram (figure 1) as additional harmonics of a new frequency ($g_0$) that is an integer fraction of $f_0$ (often $f_0/2$ or $f_0/3$). Because subharmonics literally introduce a new harmonically related tone an octave or more below $f_0$, it is not surprising that this effect was confirmed in psychoacoustic studies: subharmonics indeed lower perceived pitch, until in the limiting case only the lower frequency $g_0$ is perceived [36,37]. Like subharmonics, amplitude modulation introduces a new low-frequency component caused by the vibration of additional supra-glottal oscillators, such as flaps of soft tissue above the larynx [38], producing sidebands on the spectrogram around each harmonic of $f_0$ (figure 1). Unless these additional oscillators vibrate too slowly for the modulation frequency to create a strong pitch percept [7], the resulting pitch might again be a blend of the two frequencies, in which case amplitude modulation would lower perceived pitch similarly to the effect of subharmonics. Finally, deterministic chaos (herein, chaos) is a particularly interesting and intense case of NLP. It is caused by the vocal folds vibrating in a highly

irregular manner, as in rough barks or roars, which has the effect of 'smudging' the harmonics and introducing variable amounts of broadband noise to the spectrogram (figure 1). The resulting perceptual effect is pure harshness. Importantly, because chaos emphasizes resonance frequencies (formants), it might make it easier for listeners to estimate the length of the caller's vocal tract [39,40] and thus body size, although this has not yet been tested directly. In addition to creating a particularly harsh or rough voice quality, chaos redistributes acoustic energy in the spectrum, shifting it either up or down depending on the location of $f_0$ relative to the first formant (see Methods and figure 1). The possible effect of chaos on perceived pitch is, therefore, difficult to predict theoretically. Here, we test it empirically.

## (b) This study

To the best of our knowledge, there is no previous empirical work testing whether amplitude modulation and chaos affect perceived voice pitch, nor whether any NLP affect perceived body size. To address these important questions, we capitalized on the newly available technology for synthesizing natural-sounding vocalizations with a very precisely controlled voice quality, making it possible to create convincing examples of vocalizations with or without NLP (amplitude modulation, subharmonics and chaos) while keeping constant all other acoustic characteristics such as pitch and formants [41]. We resynthesized human non-verbal vocalizations, such as screams and roars, obtained from social media and recorded in real-life interactions and now serving as a validated corpus of genuine human vocalizations [42]. These vocal sounds are extremely rich in NLP, providing us with ecologically valid prototypes and making it possible to create our stimuli with realistic acoustic properties.

In the main playback experiment, we compared listeners' perceptions of otherwise identical resynthesized versions of human vocalizations that differed only in the presence or absence and type of NLP (figure 1). While the effect of NLP on perceived voice pitch was our main psychoacoustic outcome of interest, changes in tone brightness, rather than pitch per se, may also contribute to the perception of physical largeness, especially in the case of deterministic chaos. Further, roughness is another important psychoacoustic characteristic that increases in the presence of NLP and effectively conveys aversive affective states such as fear [28] and aggression [33]. Therefore, we included three psychoacoustic rating scales: pitch, timbre and roughness. To test how these acoustic changes modify listeners' perceptions of the speaker, we used three ecological rating scales: height, aggression and formidability. We examined perceptions of aggression and formidability separately because perceived aggressive intent might be distinct from general fighting ability or physical formidability in terms of how it is affected by voice characteristics such as $f_0$ [43].

To test in a more robust manner the key effect—whether NLP can indeed exaggerate apparent body size—we replicated the results of the rating experiment using two-alternative approaches. First, height judgements of two manipulated versions of the same vocalization were compared in a two-alternative forced choice task to tap into an explicit understanding of whether NLP-related changes of voice quality caused the speaker to sound larger or smaller. Second, we used an implicit associations test—a popular method for studying cross-modal correspondences that may exist outside of conscious awareness [44,45]. Subjects in this test are asked to classify four stimuli into two categories based on repeatedly changing pairing rules. Responses tend to be more rapid and accurate when two stimuli assigned to the same category are naturally associated. With this method, we tested whether the effect of NLP on size estimates would be observed if participants were not explicitly asked to make size judgements, and thus whether it would be likely to occur in ecologically relevant confrontational contexts that involve a rapid, and probably largely implicit, evaluation of the opponent's size. Taken together, the results of these three experiments provide convincing evidence that NLP make the voice sound lower, darker and rougher, which exaggerates the speaker's apparent size and also conveys aggression.

## 2. Methods

### (a) Stimuli

We selected 82 vocalizations with NLP from a corpus of authentic non-verbal vocalizations compiled from online sources [42]. All selected vocalizations were relatively short ($M \pm$ s.d. = $1 \pm 0.5$ s) and free from background noise, and all contained perceptually salient and manually annotated NLP. Acoustically, the stimuli consisted of intense vocalizations (screams, yells, roars, groans, etc.) and expressed various emotions such as anger, pain, fear, jubilation and surprise [42]. Each of the 82 prototype vocalizations was resynthesized in four conditions (no NLP, amplitude modulation [AM], subharmonics and chaos), for a total of $4 \times 82$ or 328 stimuli.

Highly accurate contours of fundamental frequency ($f_0$) were manually extracted with a step of 10 ms. The sounds were resynthesized with R package *soundgen* [41] using the pitch contour and smoothed spectral envelope (formants) of the original recording, but with a fully controlled glottal excitation source. This enabled us to create otherwise identical sounds with the controlled addition of amplitude modulation, subharmonics, chaos or without any NLP (figure 1). The timing and duration of NLP were matched to that of the original recording. The spectro-temporal characteristics (i.e. the frequency and depth of amplitude modulation and subharmonics, as well as the depth of chaos) were chosen randomly from the range of values previously observed to provide a good approximation of NLP in human non-verbal vocalizations [33,41,46]. All audio stimuli and R code for their synthesis are available for download at http://cogsci.se/publications.html.

### (b) Procedure

A total of 677 participants took part in the three psychoacoustic playback experiments, which were written in html/javascript and conducted online, as follows:

— *Rating task* ($n = 301$ participants). We obtained listeners' explicit ratings of 328 stimuli on three psychoacoustic (pitch, timbre, roughness) and three ecological (height, formidability, aggression) response scales in an online experiment. Each participant rated 50 stimuli in two blocks, one block with a randomly selected psychoacoustic scale and one block with a randomly selected ecological scale. The order of blocks and trials within blocks was randomized for each participant, under the constraint that the same stimulus never occurred twice. Responses were recorded on horizontal visual analogue scales ranging from 0 to 100 for judgements of timbre, roughness, formidability and aggression, from 128 cm to 202 cm for height judgements, and on a musical or logarithmic scale from 62 Hz to 4300 Hz for pitch judgements

(six octaves from C2 to C8). Two pilot tests with 20 participants each were performed to ensure that the visual appearance of rating scales, the ranges of pitch and height values, and the wording of the vignettes were appropriate. For full details on the rating scales, please see electronic supplementary material, figure S1.

— *Two-alternative forced choice task* (*n* = 192 participants). On each trial, participants were presented with two otherwise identical vocalizations that differed only in the type of NLP. Participants indicated which of the two speakers sounded taller, with an option to rate both as similar in height. Each participant completed 47 experimental trials and three catch trials with a pair of identical sounds, which were included as attention checks.

— *Implicit associations test* (*n* = 184 participants). As this method is designed to test implicit associations [44], participants did not make explicit size judgements. Rather, they were trained to press different keys in response to pairs of visual and auditory cues that were either congruent or incongruent. Because only a single pair of vocalizations could be compared at a time, testing all 82 prototypes was not feasible. Instead, we selected two vocalizations from each NLP condition (subharmonics, amplitude modulation and chaos) and paired them with the corresponding version without any NLP. The test must be fast-paced, and the stimuli must be easy to distinguish [44,45]; therefore, we chose the shortest vocalizations with strong NLP, preferring the prototypes which also displayed a large NLP effect on perceived body size in the rating experiment. We implemented a web-based version of the implicit associations test as described by Parise & Spence [44]. Listeners were required to learn a rule associating the left arrow on a keyboard or touchscreen with one image and sound, and the right arrow with another image and sound. The measure of interest was the difference in accuracy and response time in blocks with congruent (e.g. subharmonics = large) versus incongruent (e.g. subharmonics = small) combinations of images and sounds (see electronic supplementary material for more details).

## (c) Participants

All participants were recruited on the online testing platform Prolific (https://www.prolific.co/) and compensated for their time. Out of the total of 677 included participants, 428 self-identified as male, 246 as female and three as 'other'; the mean age was 26 ± 8 years (*M* ± s.d., range 18–61). Sample sizes were chosen to ensure sufficient precision of estimates of effect sizes in Bayesian multilevel models. Thus, with a sample size of 301 in the rating experiment, each of 328 experimental stimuli was rated on average 15.2 times on each of the six response scales, providing sufficient precision on the estimates of NLP effects both at population level and for each individual prototype. We recruited 200 participants for the two-alternative forced choice task. Eight failed attention checks and were thus excluded; the remaining 192 participants rated each of the 492 pairs of stimuli (82 prototypes × 6 NLP contrasts = 492 non-catch pairs of different stimuli) on average approximately 20 times, again providing sufficient precision effect size estimates. Finally, for the implicit associations test, we recruited 184 participants (6 experimental conditions × approximately 30 participants). We previously ran a similar experiment with 20 participants per condition; this provided acceptable but not particularly high precision of estimates for both accuracy and response times [45]. To make the results more robust to individual differences, in this study we increased the number of participants per condition to 30. All participants achieved high accuracy, indicating that they had understood the instructions and had no difficulty

distinguishing between pairs of stimuli; accordingly, all 184 participants were included in the analysis of implicit associations.

## (d) Data analysis

Unaggregated, trial-level data from all three experiments were analysed using Bayesian multilevel models, fit with the R package *brms* [47] with default, mildly conservative priors; for data-heavy models, these priors improve model convergence without notably affecting posterior estimates. Posterior distributions of model parameters and fitted values were summarized by their medians and 95% credible intervals (CIs). We compared credible values of effect sizes: when the credible intervals on estimates are far from the null value (e.g. zero), this indicates a credible effect given the observed data, model structure and prior knowledge. Full details of data analysis, as well as R code, are available in the electronic supplementary material.

## 3. Results

As a measure of inter-rater agreement in the rating task, we aggregated the ratings of each vocal stimulus on each response scale and calculated the mean Pearson's correlation between the responses of each participant and these aggregated ratings. These correlations were notably higher for the three psychoacoustic scales (pitch, timbre, roughness: $r = 0.74$–$0.85$) than for the ecological scales (height, formidability, aggression: $r = 0.53$–$0.60$). Likewise, the intraclass correlation coefficient, estimated using a two-way random model and absolute agreement, revealed lower reliability for the ecological scales (less than 0.2) compared to the psychoacoustic scales (0.3–0.6). The higher degree of inter-rater agreement for psychoacoustic scales indicates that these low-level, relatively objective voice properties are perceived similarly across all listeners, whereas speaker characteristics like height and aggression are perceived in a more individually variable fashion.

Effect sizes are presented as percentage points (%) for scales in which participants rated stimuli from 0 to 100 (timbre, roughness, formidability and aggression), and in natural units for height judgements (rated in cm) and pitch judgements (rated in Hz). To begin with results for the three psychoacoustic scales, all three NLP types (figure 1) lowered perceived voice pitch, made vocal timbre darker and increased the level of perceived vocal roughness in the rating task (figure 2). The effect of chaos was most pronounced: its addition to synthesized vocalizations lowered perceived pitch by 4.2 semitones (95% CI [2.8, 5.6]), caused timbre to be perceived as 11.2% [8.8, 13.7] darker and enhanced perceived roughness by 23.6% [20.5, 26.7]. The effects of subharmonics and amplitude modulation were similar to one another: both lowered perceived voice pitch by an average of 2–3 semitones (subharmonics 2.8 semitones [1.5, 4.2], amplitude modulation 2.2 semitones [0.9, 3.5]), caused the timbre to sound approximately 5% darker (subharmonics 6.1% [4.1, 8.1], amplitude modulation 5.3% [3.2, 7.4]) and increased perceived roughness by approximately 10% (subharmonics 10.5% [7.9, 13.2], amplitude modulation 13.8% [11.0, 16.5]). To assess whether the three psychoacoustic scales measured separate phenomena, we calculated the correlation between the average ratings of each vocalization on these three scales. Pitch and timbre scales were strongly correlated (Pearson's $r = 0.94$), and roughness moderately correlated with both pitch ($r = -0.69$) and timbre ($r = -0.78$).

As for the three ecological scales, all NLP had a small, but statistically robust positive effect on apparent speaker height:

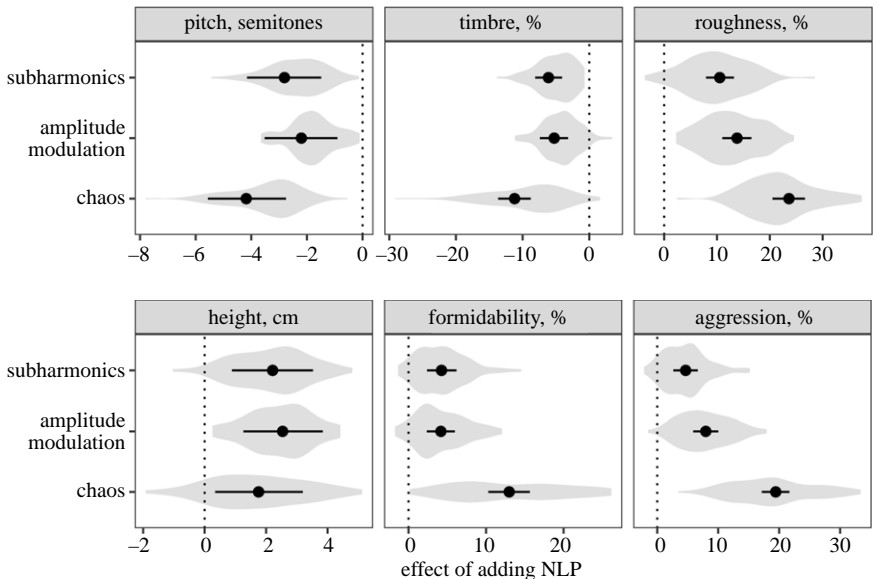

**Figure 2.** The effect of manipulating nonlinear phenomena (NLP) on listeners' ratings of vocalizations on three psychoacoustic scales (top) and three ecological scales (bottom). Solid circles show the median of posterior distribution of the difference between each condition and the no-NLP condition, with 95% CI. Violin plots show the distribution of fitted values for each prototype vocalization (N = 82). The dotted line in each panel marks null effect (no difference between sounds with and without NLP).

vocalizers sounded 1.8 cm taller [0.3, 3.2] when chaos was experimentally added to a vocalization, 2.2 cm taller [0.9, 3.5] when subharmonics were added and 2.5 cm taller [1.3, 3.8] when amplitude modulation was added. In addition, all three NLP enhanced the perceived formidability and level of aggression communicated by a given vocalization. Chaos had the greatest effect (formidability 13.0% [10.3, 15.7], aggression 19.4% [17.2, 21.7]). The effects of subharmonics and amplitude modulation on formidability were similar in magnitude (4.3% [2.4, 6.2] and 4.2% [2.4, 6.0], respectively). However, the addition of amplitude modulation increased perceived aggression twice as much (by 8.0% on average [5.9, 10.0]) as did the addition of subharmonics (4.7% [2.7, 6.7]). Aggression and formidability scales were moderately correlated (r = 0.70), while the height scale was relatively independent (r = 0.47 with formidability and r = 0.31 with pitch), suggesting that physical size and physical formidability are related but not identical constructs.

When presented with two otherwise identical vocalizations that only differed in NLP, listeners reliably indicated that vocalizations with experimentally added chaos were produced by a taller person. The vocalization with chaos was chosen as originating from the larger vocalizer 38.0% [28.4, 47.4] of the time compared to vocalizations without NLP, 29.3% of the time [21.4, 37.3] compared to amplitude modulation and 24.9% [18.3, 31.6] compared to subharmonics (figure 3a). The effect of subharmonics and amplitude modulation relative to each other and relative to the absence of any NLP was not statistically different from noise level (defined as the difference between catch pairs of identical vocalizations). Furthermore, there was high variability among the 82 prototype vocalizations in terms of the effect of chaos. For example, comparing chaos to no NLP, the difference was not statistically different from noise for four out of 82 prototypes, and for three of 82 prototypes the effect of chaos was reversed (violin plots in figure 3a). In sum, the effect of NLP on perceived speaker size was less robust when listeners' attention was explicitly drawn to the nature

of acoustic manipulation, with only deterministic chaos being reliably associated with large size.

By contrast, for the implicit associations test, statistically credible congruency effects were observed for all three NLP types in five out of six tested sounds, for which 95% CIs for both errors and response time excluded the null value (figure 3b). The results illustrate that, in a pair of two otherwise identical vocalizations, participants consistently found it more natural to associate a vocalization with NLP with a large person, and a vocalization without NLP with a small person.

# 4. Discussion

As described by Morton nearly 50 years ago [13] and confirmed in numerous later studies (reviewed in [30,35]), a rough or harsh voice quality associated with NLP is often found in aggressive calls of animals, and is also perceived as aversive in human vocalizations [33,34]. To explain the function of this harsh vocal quality, it has been hypothesized that NLP lower perceived voice pitch and therefore make the vocalizer appear larger [17], with obvious potential benefits for vocalizers. Indeed, such a mechanism for size exaggeration could be adaptive in aggressive contexts such as competitive or dominance displays and mating contests. Until recently, however, it was impossible to directly manipulate NLP in natural-sounding calls to test their effect on perceived body size or aggressive intent. Here, we took advantage of recent advances in parametric voice synthesis to put this hypothesis to the test, separately evaluating the effect of three NLP types—subharmonics, amplitude modulation and chaos—on perceived pitch and speaker size in human non-verbal vocalizations including roars, screams, groans and moans.

The results of our perceptual experiments clearly show that all tested NLP types indeed lower the apparent voice pitch and increase the apparent size of the vocalizer. Furthermore, in a replication using the implicit associations test, we show that the effect of NLP on perceived size is not confined

(a) two-alternative forced choice

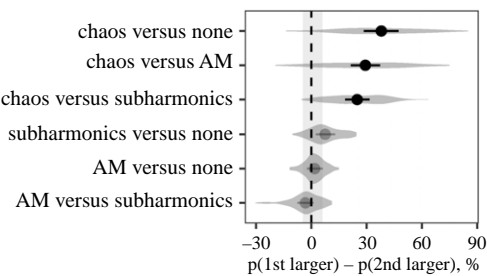

(b) implicit associations test

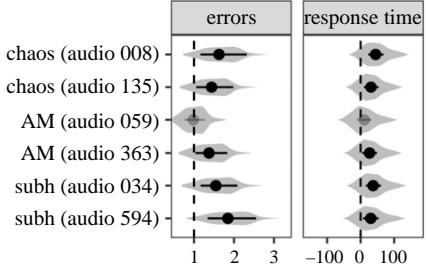

**Figure 3.** (a) The effect of manipulating nonlinear phenomena (NLP) on perceived body size in the two-alternative forced choice task: fitted values from ordinal logistic regression. Markers indicate the probability of perceiving the first vocalization (e.g. in the top example, a vocalization with chaos) as larger than the second vocalization (e.g. a vocalization without any NLP, 'none') minus the probability of perceiving it as smaller (median of posterior distribution and 95% CI), ignoring ties. If the point lies to the right of the dotted line, it means that the first listed NLP condition was associated with larger size; 100% = all listeners always chose the first vocalization. The greyed-out markers have CIs that fail to clear the region of practical equivalence corresponding to the effect size for catch pairs of perfectly identical sounds. (b) Implicit associations tests showing the increase in errors (odds ratio, left panel) and response times (ms, right panel) in blocks with incongruent versus congruent combinations, where a congruent combination is defined as pairing a sound with NLP with the image of a large person, and a sound without NLP with the image of a small person. The solid points show population-level effects (medians of posterior distribution and 95% CI, greyed out if the CI fails to clear the null value), and the violin plots show the distribution of fitted values across participants.

to explicit size judgements. Equally important, we created fully realistic vocalizations, with the same nonlinear acoustic characteristics as naturally observed exemplars. In other words, the kind of NLP that naturally occur in human non-verbal vocalizations, such as screams and angry roars, lower perceived voice pitch and serve to exaggerate the apparent size of the caller. Because the anatomical mechanisms of vocal production [40] and perception [48] are broadly similar among humans and other terrestrial mammals, it is likely that these results can be generalized to vocal communication in non-human mammals, a hypothesis that can now be directly tested.

In addition to exaggerating speaker size, we show that NLP also increase perceived aggression. Interestingly, while all NLP were equally effective at size exaggeration, chaos was perceived as both rougher and more aggressive than either subharmonics or amplitude modulation. In our earlier studies, we also found a similar distinction among NLP types, with chaos perceived as most aversive [33,46]. This distinction indicates that listeners can discriminate between a person who sounds impressively large versus a person who sounds aggressive, complementing recent work showing that aggressive intent and fighting ability are deferentially signalled by voice pitch lowering in men [43]. Our result also emphasizes the need to annotate different NLP types separately when analysing voice recordings. Subharmonics and sidebands are notoriously difficult to distinguish even for trained bioacousticians, and their perceptual effects appear to be comparable, but deterministic chaos is easily distinguishable and appears to elicit very different perceptual effects.

While our results show that NLP lower perceived voice pitch, we cannot be certain that NLP exaggerate the caller's body size *because* of this pitch-lowering effect. Indeed, the addition of NLP in our study also caused voice timbre to be perceived as darker. Another consequence of adding NLP was an increase in vocal roughness, based on both subjective listeners' ratings and objective acoustic measurements. Some combination of these and other psychoacoustic consequences of the presence of NLP may contribute to the observed effect on apparent body size. However, it is very well established that low-pitched voices project the impression of a large body size [9,12], so the pitch-lowering effect of NLP is certain

to contribute to size exaggeration, even if it may not be solely responsible for it.

Methodologically, a comparison of the outcome of the three alternative experimental designs for measuring the effect of NLP on perceived size yielded valuable insight regarding the respective strengths and limitations of these approaches. Belying the intuition, and previous reports [49], that the most explicit tasks should amplify experimental effects, only chaos was observed to enhance perceived size in a two-alternative forced choice task, in which participants were both explicitly aware of making size judgements and encouraged to note the nature of acoustic manipulation. By contrast, all NLP had a robust effect on perceived size when one vocalization at a time was rated on a visual analogue scale. This between-subject design can be seen as intermediate in explicitness because participants were asked to make size judgements, but were less likely to guess the nature of manipulation as the same listener never rated two versions of the same prototype vocalization. By contrast, the implicit associations test gave away the nature of acoustic manipulation, but demanded no explicit size judgements. We observed excellent convergence between the results of the implicit associations test and size ratings on an analogue scale, but investigating implicit associations requires considerably more data. Based on our results, we would, therefore, cautiously recommend using ratings on a continuous scale in a between-subject design as a sensitive and resource-efficient method for measuring the effect of acoustic manipulations in human vocal stimuli.

In conclusion, the association of NLP with body size and formidability demonstrated in this work sheds new light on the communicative role and evolution of irregular phonation. While harsh-sounding NLP may constitute regrettable voice imperfections in the context of speech therapy or classical singing, biologists have increasingly recognized their potential for attracting and holding attention [31,32], communicating high arousal [17,29,30] and even attracting mates [24]. If we add the size exaggerating effect in agonistic interactions, it is easy to see how selection may favour anatomical structures and physiological mechanisms that render the vocal organ unstable and capable of supporting complex vibratory regimes such as subharmonics and chaos, as indeed observed in the vocalizations of a wide range of animals, from birds and frogs to

dogs and whales [17,20,21,23,24,29,31]. From the receivers' perspective, the ability of NLP to prevent habituation is presumably based on their unpredictable nature and is thus derived from attentional mechanisms, while the present findings indicate that their size exaggeration effect is mediated by a cross-modal correspondence between size and auditory frequency or pitch. An important task in unravelling the evolution of vocal communication will be to achieve a fuller understanding of the nature and origins of such cognitive mechanisms, which may ultimately explain many universal features found in the acoustic code of different species.

Ethics. All participants provided informed consent. Ethical approval for performing perceptual experiments with human subjects was provided by the Comité d'Ethique du CHU de Saint-Etienne (IRBN692019/CHUSTE).

Data accessibility. All datasets, R scripts for data analysis, audio stimuli, R scripts for generating synthesized stimuli and html code for running psychoacoustic experiments can be downloaded from http://cogsci.se/publications.html. These supplemental materials enable full validation and replication of results.

The data are provided in the electronic supplementary material [50].

Authors' contributions. A.A.: conceptualization, formal analysis, writing—original draft; K.P.: conceptualization, writing—review and editing; M.M.: conceptualization, writing—review and editing; D.R.: conceptualization, writing—review and editing.

All authors gave final approval for publication and agreed to be held accountable for the work performed therein.

Competing interests. The authors declare no competing interests.

Funding. A.A. was supported by grant 2020-06352 from the Swedish Research Council. K.P., M.M. and D.R. were supported by the University of Lyon IDEXLYON project as part of the 'Programme Investissements d'Avenir' (ANR-16-IDEX-0005) to D.R.

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
