## [Peer Review File · Proceedings of the Royal Society B: Biological Sciences]

Review History

RSPB-2021-0872.R0 (Original submission)

Review form: Reviewer 1 (Daniel Blumstein)

Recommendation

Accept as is

Scientific importance: Is the manuscript an original and important contribution to its field?

Excellent

General interest: Is the paper of sufficient general interest?

Excellent

Quality of the paper: Is the overall quality of the paper suitable?

Excellent

Is the length of the paper justified?

Yes

Should the paper be seen by a specialist statistical reviewer?

No

Do you have any concerns about statistical analyses in this paper? If so, please specify them explicitly in your report.

No

It is a condition of publication that authors make their supporting data, code and materials available - either as supplementary material or hosted in an external repository. Please rate, if applicable, the supporting data on the following criteria.

Is it accessible?

Yes

Is it clear?

Yes

Is it adequate?

Yes

Do you have any ethical concerns with this paper?

No

Comments to the Author

Review of: Anikin et al. Harsh is large

Review by: Dan Blumstein

The authors manipulated the structure of human nonverbal signals by adding specific types of non-linear vocal phenomena to them and found that these modified sounds were perceived as coming from larger, and more aggressive and formidable signalers as well as being perceived as lower pitched, rougher, and having reduced timber. I loved this paper and think that it was well-conceived and conducted.

My questions are entirely about methodological details.

Not knowing about how the synthesis program actually works, I wanted more details about synthesis. Adding 'chaos' synthetically can create sounds that are 'clicky' or 'noisy'--these clearly were noisy. What sort of parameters were used to synthesize sounds? This would ensure reproducibility. I eventually found more details in the supplement and all details are in the R code but suggest this must be better referred to in the main text (more than on line 197).

I wanted to have more information on how participants actually rated sounds. What prompts were they given? How was aggression defined? How was formidability defined? Etc. Again, I eventually found details in the supplementary materials...referencing in the main text is essential.

L112: there's something wrong here...

Review form: Reviewer 2

Recommendation

Accept with minor revision (please list in comments)

Scientific importance: Is the manuscript an original and important contribution to its field?

Excellent

General interest: Is the paper of sufficient general interest?

Excellent

Quality of the paper: Is the overall quality of the paper suitable?

Excellent

Is the length of the paper justified?

Yes

Should the paper be seen by a specialist statistical reviewer?

No

Do you have any concerns about statistical analyses in this paper? If so, please specify them explicitly in your report.

No

It is a condition of publication that authors make their supporting data, code and materials available - either as supplementary material or hosted in an external repository. Please rate, if applicable, the supporting data on the following criteria.

Is it accessible?

No

Is it clear?

Yes

Is it adequate?

Yes

Do you have any ethical concerns with this paper?

No

Comments to the Author

This was a very interesting manuscript to read that provides for an important test of some long-held questions about what information might be encoded into animal vocalizations in various types of nonlinear phenomena (NLP) and how receivers might decode that information. In my view, the authors have applied some cutting-edge techniques to address these questions; they have done so in a rigorous fashion; and the conclusions conservatively follow directly from the clear results of Bayesian analyses. My specific comments below are simply intended to help the authors consider some additional things in crafting the very best revision possible.

Line 59: The authors casually mention "roars" here and in several other places in the MS as being a typical human vocalization. But I don't think most readers will, in fact, agree that "roars" as being human vocalizations. Yells, screams, cries...yes, all human. But roars? When I think of instances when I've heard a human "roar," they were invariably imitating other mammals that do roar, like lions and tigers, etc. I think some additional explanation about what the authors mean when they say that humans roar would help include more readers in the narrative.

Line 68: What about "and even frogs" here? If you want to cast the broadest net possible in your narrative, I suggest citing some of the several papers that have now documented NLPs in frog vocalizations:

Suthers, R. A., Narins, P. M., Lin, W. Y., Schnitzler, H. U., Denzinger, A., Xu, C. H., & Feng, A. S. (2006). Voices of the dead: Complex nonlinear vocal signals from the larynx of an ultrasonic frog. *Journal of Experimental Biology*, 209(24), 4984-4993.

Feng, A. S., & Narins, P. M. (2008). Ultrasonic communication in concave-eared torrent frogs (*Amolops tormotus*). *Journal of Comparative Physiology A*, 194(2), 159-167.

Feng, A. S., Riede, T., Arch, V. S., Yu, Z. L., Xu, Z. M., Yu, X. J., & Shen, J. X. (2009). Diversity of the vocal signals of concave-eared torrent frogs (*Odorrana tormota*): Evidence for individual signatures. *Ethology*, 115(11), 1015-1028.

Pettitt, B. A., Bourne, G. R., & Bee, M. A. (2012). Quantitative acoustic analysis of the vocal repertoire of the golden rocket frog (*Anomaloglossus beebei*). *Journal of the Acoustical Society of America*, 131, 4811-4820.

Zhang, F., Zhao, J., & Feng, A. S. (2017). Vocalizations of female frogs contain nonlinear characteristics and individual signatures. *Plos One*, 12(3).

Zhang, F., Wu, Y. T., Wang, J. M., & Bao, J. H. (2021). Calls of the Large Odorous Frog (*Odorrana graminea*) Show Nonlinear and Individual Characteristics. *Asian Herpetological Research*, 12(1), 124-134.

Wu, Y. T., Bao, J. H., Lee, P. S., Wang, J. M., Wang, S., & Zhang, F. (2021). Nonlinear Phenomena Conveying Body Size Information and Improving Attractiveness of the Courtship Calls in the Males of *Odorrana tormota*. *Asian Herpetological Research*, 12(1), 117-123.

Line 112: Is something wrong here with the indication concerning from where sounds were obtained?

Lines 126-135: A lot of this information fell like it would have been more appropriate in the Introduction than in the first paragraph of the Methods.

Line 136: The manuscript seems to get a little ahead of itself here. It would seem better to me to describe taking a robust approach to testing your hypothesis, and not getting a result and then testing to see if it was robust. It's a subtly wording difference, perhaps, but I think describing your approach as robust in the methods and not your "key finding" (which should not be reported in the Methods anyway) would make for a stronger manuscript.

Line 140: Consider adding one more sentence, or even a partial sentence, here describing what the Implicit Associations Test entails.

Line 148 and elsewhere: Please specify what is being reported after the +/-...SD? CI?

Line 163: While the presentation of the stimulus examples on the author's personal website is very slick and professional, I think these examples need to be made available in a proper archive of some kind.

Lines 207-210: I had some trouble following the math here. For example, $82 \times 6 = 492$, not 480; $6 \times 30 = 180$, not 184.

Line 218: Can please you be more specific here in describing what you mean by "default, mildly conservative priors"?

Line 224-233: These struck me more as potentially interesting results that belong in a Results section and less material related to Methods. Might there be some cool biological reasons why inter-rater agreement would be higher on basic psychoacoustic measures than on interpretations of signal information?

Line 234: This comment applies to most of the Results section, which I read before reading the Supplementary Material (which may also be likely for future readers). I found it difficult to interpret the various percentage values reported because they did not have all the context I needed. And to be honest, I'm still unclear about what some of them are really measuring, and how comparable they are to each other. What does it mean for "timbre to be perceived as 11.2% darker"? Is that perceptually comparable to an 11.2% change in roughness? I think in general, the main text would benefit from some additional explanation about what a percentage difference in perception really means in this particular study.

Line 252: The "Once again" here seemed to refer to earlier text in this paragraph, but in that text, chaos had the smallest, not the greatest effect. So I was confused.

Line 259: Unclear to me what "related but independent constructs" means here in light of the non-trivial correlations cited just prior to this.

Figure 2 Legend: Can you clarify what is meant by "marks null effect"? Does this mean an explicit comparison to results obtained with the no-NLP stimuli?

Decision letter (RSPB-2021-0872.R0)

07-Jun-2021

Dear Dr Anikin

I am pleased to inform you that your Review manuscript RSPB-2021-0872 entitled "Harsh is large: nonlinear vocal phenomena lower voice pitch and exaggerate body size" has been accepted for publication in Proceedings B.

The referee(s) do not recommend any further changes. Therefore, please proof-read your manuscript carefully and upload your final files for publication. Because the schedule for publication is very tight, it is a condition of publication that you submit the revised version of your manuscript within 7 days. If you do not think you will be able to meet this date please let me know immediately.

To upload your manuscript, log into <http://mc.manuscriptcentral.com/prsb> and enter your Author Centre, where you will find your manuscript title listed under "Manuscripts with Decisions." Under "Actions," click on "Create a Revision." Your manuscript number has been appended to denote a revision.

You will be unable to make your revisions on the originally submitted version of the manuscript. Instead, upload a new version through your Author Centre.

- 1) A text file of the manuscript (doc, txt, rtf or tex), including the references, tables (including captions) and figure captions. Please remove any tracked changes from the text before submission. PDF files are not an accepted format for the "Main Document".
- 2) A separate electronic file of each figure (tiff, EPS or print-quality PDF preferred). The format should be produced directly from original creation package, or original software format. Please note that PowerPoint files are not accepted.

3) Electronic supplementary material: this should be contained in a separate file from the main text and the file name should contain the author's name and journal name, e.g. `authorname_procb_ESM_figures.pdf`

All supplementary materials accompanying an accepted article will be treated as in their final form. They will be published alongside the paper on the journal website and posted on the online figshare repository. Files on figshare will be made available approximately one week before the accompanying article so that the supplementary material can be attributed a unique DOI. Please see: <https://royalsociety.org/journals/authors/author-guidelines/>

4) Data-Sharing and data citation

It is a condition of publication that data supporting your paper are made available. Data should be made available either in the electronic supplementary material or through an appropriate repository. Details of how to access data should be included in your paper. Please see <https://royalsociety.org/journals/ethics-policies/data-sharing-mining/> for more details.

If you wish to submit your data to Dryad (<http://datadryad.org/>) and have not already done so you can submit your data via this link <http://datadryad.org/submit?journalID=RSPB&manu=RSPB-2021-0872> which will take you to your unique entry in the Dryad repository.

Once again, thank you for submitting your manuscript to Proceedings B and I look forward to receiving your final version. If you have any questions at all, please do not hesitate to get in touch.

Sincerely,
Dr Robert Barton
<mailto:proceedingsb@royalsociety.org>

Associate Editor
Comments to Author:

This is a well-written manuscript describing a clever and carefully designed study. My original high enthusiasm after my first assessment has only grown after reading it again and evaluating the responses of the reviewers.

It was wonderful to see similar enthusiasm from both reviewers. Reviewer 2 bring up a few, minor points that will add clarity to the manuscript. I particularly like this reviewer's suggestion to broaden the taxonomic groups for which the question posted here is relevant by including studies on frogs. I am grateful for their suggestion of references and would love to see you including this perspective in the study.

I am excited about seeing this study published at Proc B and I am looking forward to seeing studies that follow up your work and test this hypothesis in non-human systems,
Ximena Bernal

Reviewer(s)' Comments to Author:
Referee: 1
Comments to the Author(s)
Review of: Anikin et al. Harsh is large
Review by: Dan Blumstein

The authors manipulated the structure of human nonverbal signals by adding specific types of non-linear vocal phenomena to them and found that these modified sounds were perceived as coming from larger, and more aggressive and formidable signalers as well as being perceived as lower pitched, rougher, and having reduced timber. I loved this paper and think that it was well-conceived and conducted.

My questions are entirely about methodological details.

Not knowing about how the synthesis program actually works, I wanted more details about synthesis. Adding 'chaos' synthetically can create sounds that are 'clicky' or 'noisy'--these clearly were noisy. What sort of parameters were used to synthesize sounds? This would ensure reproducibility. I eventually found more details in the supplement and all details are in the R code but suggest this must be better referred to in the main text (more than on line 197).

I wanted to have more information on how participants actually rated sounds. What prompts were they given? How was aggression defined? How was formidability defined? Etc. Again, I eventually found details in the supplementary materials...referencing in the main text is essential.

L112: there's something wrong here...

Referee: 2

Comments to the Author(s)

This was a very interesting manuscript to read that provides for an important test of some long-held questions about what information might be encoded into animal vocalizations in various types of nonlinear phenomena (NLP) and how receivers might decode that information. In my view, the authors have applied some cutting-edge techniques to address these questions; they have done so in a rigorous fashion; and the conclusions conservatively follow directly from the clear results of Bayesian analyses. My specific comments below are simply intended to help the authors consider some additional things in crafting the very best revision possible.

Line 59: The authors casually mention "roars" here and in several other places in the MS as being a typical human vocalization. But I don't think most readers will, in fact, agree that "roars" as being human vocalizations. Yells, screams, cries...yes, all human. But roars? When I think of instances when I've heard a human "roar," they were invariably imitating other mammals that do roar, like lions and tigers, etc. I think some additional explanation about what the authors mean when they say that humans roar would help include more readers in the narrative.

Line 68: What about "and even frogs" here? If you want to cast the broadest net possible in your narrative, I suggest citing some of the several papers that have now documented NLPs in frog vocalizations:

Suthers, R. A., Narins, P. M., Lin, W. Y., Schnitzler, H. U., Denzinger, A., Xu, C. H., & Feng, A. S. (2006). Voices of the dead: Complex nonlinear vocal signals from the larynx of an ultrasonic frog. *Journal of Experimental Biology*, 209(24), 4984-4993.

Feng, A. S., & Narins, P. M. (2008). Ultrasonic communication in concave-eared torrent frogs (*Amolops tormotus*). *Journal of Comparative Physiology A*, 194(2), 159-167.

Feng, A. S., Riede, T., Arch, V. S., Yu, Z. L., Xu, Z. M., Yu, X. J., & Shen, J. X. (2009). Diversity of the vocal signals of concave-eared torrent frogs (*Odorrana tormota*): Evidence for individual signatures. *Ethology*, 115(11), 1015-1028.

Pettitt, B. A., Bourne, G. R., & Bee, M. A. (2012). Quantitative acoustic analysis of the vocal repertoire of the golden rocket frog (*Anomaloglossus beebei*). *Journal of the Acoustical Society of America*, 131, 4811-4820.

Zhang, F., Zhao, J., & Feng, A. S. (2017). Vocalizations of female frogs contain nonlinear characteristics and individual signatures. *Plos One*, 12(3).

Zhang, F., Wu, Y. T., Wang, J. M., & Bao, J. H. (2021). Calls of the Large Odorous Frog (*Odorrana graminea*) Show Nonlinear and Individual Characteristics. *Asian Herpetological Research*, 12(1), 124-134.

Wu, Y. T., Bao, J. H., Lee, P. S., Wang, J. M., Wang, S., & Zhang, F. (2021). Nonlinear Phenomena Conveying Body Size Information and Improving Attractiveness of the Courtship Calls in the Males of *Odorrana tormota*. *Asian Herpetological Research*, 12(1), 117-123.

Line 112: Is something wrong here with the indication concerning from where sounds were obtained?

Lines 126-135: A lot of this information fell like it would have been more appropriate in the Introduction than in the first paragraph of the Methods.

Line 136: The manuscript seems to get a little ahead of itself here. It would seem better to me to describe taking a robust approach to testing your hypothesis, and not getting a result and then testing to see if it was robust. It's a subtly wording difference, perhaps, but I think describing your approach as robust in the methods and not your "key finding" (which should not be reported in the Methods anyway) would make for a stronger manuscript.

Line 140: Consider adding one more sentence, or even a partial sentence, here describing what the Implicit Associations Test entails.

Line 148 and elsewhere: Please specify what is being reported after the +/-...SD? CI?

Line 163: While the presentation of the stimulus examples on the author's personal website is very slick and professional, I think these examples need to be made available in a proper archive of some kind.

Lines 207-210: I had some trouble following the math here. For example, $82 \times 6 = 492$, not 480; $6 \times 30 = 180$, not 184.

Line 218: Can please you be more specific here in describing what you mean by "default, mildly conservative priors"?

Line 224-233: These struck me more as potentially interesting results that belong in a Results section and less material related to Methods. Might there be some cool biological reasons why inter-rater agreement would be higher on basic psychoacoustic measures than on interpretations of signal information?

Line 234: This comment applies to most of the Results section, which I read before reading the Supplementary Material (which may also be likely for future readers). I found it difficult to interpret the various percentage values reported because they did not have all the context I needed. And to be honest, I'm still unclear about what some of them are really measuring, and how comparable they are to each other. What does it mean for "timbre to be perceived as 11.2% darker"? Is that perceptually comparable to an 11.2% change in roughness? I think in general, the main text would benefit from some additional explanation about what a percentage difference in perception really means in this particular study.

Line 252: The "Once again" here seemed to refer to earlier text in this paragraph, but in that text, chaos had the smallest, not the greatest effect. So I was confused.

Line 259: Unclear to me what “related but independent constructs” means here in light of the non-trivial correlations cited just prior to this.

Figure 2 Legend: Can you clarify what is meant by “marks null effect”? Does this mean an explicit comparison to results obtained with the no-NLP stimuli?

Author's Response to Decision Letter for (RSPB-2021-0872.R0)

See Appendix A.

Decision letter (RSPB-2021-0872.R1)

11-Jun-2021

Dear Dr Anikin

I am pleased to inform you that your manuscript entitled "Harsh is large: nonlinear vocal phenomena lower voice pitch and exaggerate body size" has been accepted for publication in Proceedings B.

Your article has been estimated as being 9 pages long. Our Production Office will be able to confirm the exact length at proof stage.

Data Accessibility section

Open Access

Paper charges

Sincerely,
Editor, Proceedings B
mailto: proceedingsb@royalsociety.org

Appendix A

Dear Dr Barton,

We are delighted to receive this positive evaluation of our work and are grateful for your helpful suggestions, as well as the valuable comments of Dr Blumstein and an anonymous second reviewer. We here present an updated version of the manuscript, in which we carefully addressed these comments, as explained in detail below. The changes are largely concerned with clarifying our methods and results and improving the readability of the manuscript: for example, we moved some parts of Methods into our Introduction and Results, provided additional details on sound synthesis and data analysis, and corrected several typos. We are particularly grateful for suggesting the references on NLPs in amphibians, which are now incorporated in the text.

We thank you warmly once again for considering our work.

With our best wishes,

Andrey Anikin, Kasia Pisanski, Mathilde Massenet, and David Reby

###

E_1 Reviewer 2 bring up a few, minor points that will add clarity to the manuscript. I particularly like this reviewer's suggestion to broaden the taxonomic groups for which the question posted here is relevant by including studies on frogs. I am grateful for their suggestion of references and would love to see you including this perspective in the study.

RESPONSE: we are glad to follow this suggestion and mention other animal groups to broaden the potential relevance and reach of the manuscript (see also comment R2_2). We included two of the suggested references to NLPs in amphibians (Suthers et al., 2006 and the particularly relevant Wu et al., 2021, who also discuss NLPs in relation to body size in frogs).

###

R1_1 Not knowing about how the synthesis program actually works, I wanted more details about synthesis. Adding 'chaos' synthetically can create sounds that are 'clicky' or 'noisy'--these clearly were noisy. What sort of parameters were used to synthesize sounds? This would ensure reproducibility. I eventually found more details in the supplement and all details are in the R code but suggest this must be better referred to in the main text (more than on line 197).

RESPONSE: We referred more consistently to supplements in the main text and added some details about sound synthesis to the caption of Figure 10:

“Amplitude modulation is created by multiplying the signal by a relatively low-frequency waveform ($M \pm SD = 90 \pm 20$ Hz); subharmonics are synthesized as a new, harmonically related voiced component at 1/2, 1/3, or 1/4 of the original fundamental frequency; chaos is emulated by adding strong jitter – rapid random pitch changes (audio and code for creating the stimuli are in supplements).”

R1_2 I wanted to have more information on how participants actually rated sounds. What prompts were they given? How was aggression defined? How was formidability defined? Etc. Again, I eventually found details in the supplementary materials...referencing in the main text is essential.

RESPONSE: We moved the details on rating scales from supplements to the main text (Methods / (a) Stimuli):

“Responses were recorded on horizontal Visual Analog Scales ranging from 0 to 100 for judgments of timbre, roughness, formidability, and aggression, from 128 cm to 202 cm for height judgments, and on a musical or logarithmic scale from 62 Hz to 4300 Hz for pitch judgments (six octaves from C2 to C8, easily covering the range of human voices). Two pilot tests with 20 participants each were performed to ensure that the visual appearance of rating scales, the ranges of pitch and height values, and the wording of the vignettes were appropriate. For full details on the rating scales, please see supplementary Fig S1).”

R1_3 L112: there’s something wrong here...

RESPONSE: this looks like a glitch at pdf conversion stage, it’s not in the text.

###

R2_1 Line 59: The authors casually mention “roars” here and in several other places in the MS as being a typical human vocalization. But I don’t think most readers will, in fact, agree that “roars” as being human vocalizations. Yells, screams, cries...yes, all human. But roars? When I think of instances when I’ve heard a human “roar,” they were invariably imitating other mammals that do roar, like lions and tigers, etc. I think some additional explanation about what the authors mean when they say that humans roar would help include more readers in the narrative.

RESPONSE: The English word “roar”, although admittedly more common in reference to animal sounds, is both well-established in the literature on human nonverbal vocalizations and used spontaneously by human listeners asked to classify these vocalizations (Anikin, A., Bååth, R., & Persson, T. (2018). Human non-linguistic vocal repertoire: call types and their meaning. *Journal of Nonverbal Behavior*, 42(1), 53-80.)

R2_2 Line 68: What about “and even frogs” here? If you want to cast the broadest net possible in your narrative, I suggest citing some of the several papers that have now documented NLPs in frog vocalizations

RESPONSE: Thank you for this suggestion! We have added two of the suggested references both here and in Discussion.

R2_3 Line 112: Is something wrong here with the indication concerning from where sounds were obtained?

RESPONSE: see R1_3 (formatting glitch).

R2_4 Lines 126-135: A lot of this information fell like it would have been more appropriate in the Introduction than in the first paragraph of the Methods.

RESPONSE: We created a new subsection in our Introduction (“This study”), which incorporated this information with some minor reshuffling of the text.

R2_5 Line 136: The manuscript seems to get a little ahead of itself here. It would seem better to me to describe taking a robust approach to testing your hypothesis, and not getting a result and then testing to see if it was robust. It’s a subtly wording difference, perhaps, but I think describing your approach as robust in the methods and not your “key finding” (which should not be reported in the Methods anyway) would make for a stronger manuscript.

RESPONSE: Changed to:

“To test in a more robust manner the key effect – whether NLPs can indeed exaggerate apparent body size – we replicated the results of the rating experiment using two alternative approaches.”

R2_6 Line 140: Consider adding one more sentence, or even a partial sentence, here describing what the Implicit Associations Test entails.

RESPONSE: Added text:

“Subjects in this test are asked to classify four stimuli into two categories based on repeatedly changing pairing rules. Responses tend to be more rapid and accurate when two stimuli assigned to the same category are naturally associated.”

R2_7 Line 148 and elsewhere: Please specify what is being reported after the \pm ...SD? CI?

RESPONSE: corrected, thank you! $M \pm SD$

R2_8 Line 163: While the presentation of the stimulus examples on the author’s personal website is very slick and professional, I think these examples need to be made available in a proper archive of some kind.

RESPONSE: We believe that the most user-friendly way to present a few examples of our stimuli is in an html document with spectrograms and audio next to them, rather than in an archived form.

R2_9 Lines 207-210: I had some trouble following the math here. For example, $82 \times 6 = 492$, not 480; $6 \times 30 = 180$, not 184.

RESPONSE: $82 \times 6 = 492$ has been corrected, thank you! 6×30 is not a mistake: because of how the recruiting platform Prolific works, the number of participants is sometimes a little different from the target number. So in this case, we aimed to include 30 per condition, but in reality we got four extra, so $180 + 4 = 184$. New text:

“Eight failed attention checks and were thus excluded; the remaining 192 participants rated each of the 492 pairs of stimuli (82 prototypes $\times 6$ NLP contrasts = 492 non-catch pairs of different stimuli) on average approximately 20 times, again providing sufficient precision effect size estimates. Finally, for the Implicit Associations Test, we recruited 184 participants (6 experimental conditions \times approximately 30 participants).”

R2_10 Line 218: Can please you be more specific here in describing what you mean by “default, mildly conservative priors”?

RESPONSE: A full discussion is really beyond the scope of this paper. We refer an interested reader to the excellent book by McElreath (2020) “A Bayesian rethinking”, 2nd ed. New text:

“with default, mildly conservative priors; for models with so much data, these priors simply improve model convergence without noticeably affecting the posterior estimates.”

R2_11 Line 224-233: These struck me more as potentially interesting results that belong in a Results section and less material related to Methods. Might there be some cool biological reasons why inter-rater agreement would be higher on basic psychoacoustic measures than on interpretations of signal information?

RESPONSE: We agree, so this paragraph now opens the Results section. As for the reason for higher agreement on psychoacoustic measures, this seems to be a consequence of pitch / timbre / roughness being basic, “objective” sound properties. In fact, the corresponding ratings can be mathematically predicted with high accuracy from a model of basic auditory perception, which is what the discipline of psychoacoustics is all about. In contrast, speaker characteristics, such as height or aggression, are presumably only indirectly derived from these low-level sound qualities and may be affected by many other idiosyncratic factors such as mood, personal associations, etc.

R2_12 Line 234: This comment applies to most of the Results section, which I read before reading the Supplementary Material (which may also be likely for future readers). I found it difficult to interpret

the various percentage values reported because they did not have all the context I needed. And to be honest, I'm still unclear about what some of them are really measuring, and how comparable they are to each other. What does it mean for "timbre to be perceived as 11.2% darker"? Is that perceptually comparable to an 11.2% change in roughness? I think in general, the main text would benefit from some additional explanation about what a percentage difference in perception really means in this particular study.

RESPONSE: To make the reported effect sizes easier to understand, we added this text (the 2nd paragraph of the new Results section):

"Effect sizes are presented as percentage points (%) for the four rating scales measured in arbitrary units from 0 to 100 (timbre, roughness, formidability, and aggression), and in natural units for height and pitch. To begin with the three psychoacoustic scales, all three NLP types..."

R2_13 Line 252: The "Once again" here seemed to refer to earlier text in this paragraph, but in that text, chaos had the smallest, not the greatest effect. So I was confused.

RESPONSE: Thank you for pointing this out, we have removed the "once again"!

R2_14 Line 259: Unclear to me what "related but independent constructs" means here in light of the non-trivial correlations cited just prior to this.

RESPONSE: Corrected to:

"suggesting that physical size and physical formidability are related but not identical constructs"

R2_15 Figure 2 Legend: Can you clarify what is meant by "marks null effect"? Does this mean an explicit comparison to results obtained with the no-NLP stimuli?

RESPONSE: Thank you for this important comment! To improve the transparency of our main results in Fig. 2 and in the text, we extended the caption:

"Solid circles show the median of posterior distribution of the difference between each condition and the no-NLP condition, with 95% CI. Violin plots show the distribution of fitted values for each prototype vocalization (N = 82). The dotted line in each panel marks null effect (no difference between sounds with and without NLPs)."

We also clarified how these contrasts were calculated in Supplements / Data analysis / Rating test:

"The effect sizes shown in Figure 2 in the main text are contrasts between the ratings of sounds with and without NLPs: for each step in the MCMC chain, we calculated the difference in fitted values between two conditions and then summarized the resulting posterior distribution by its median and 95% CI."